# Transmission of Raccoon-Passaged Chronic Wasting Disease Agent to White-Tailed Deer

**DOI:** 10.3390/v14071578

**Published:** 2022-07-20

**Authors:** Eric D. Cassmann, Alexis J. Frese, S. Jo Moore, Justin J. Greenlee

**Affiliations:** 1Virus and Prion Research Unit, National Animal Disease Center, Agricultural Research Service, United States Department of Agriculture, Ames, IA 50010, USA; eric.cassmann@usda.gov (E.D.C.); alexis.frese@usda.gov (A.J.F.); jo.moore@moredun.ac.uk (S.J.M.); 2Oak Ridge Institute for Science and Education, 1299 Bethel Valley Rd., Oak Ridge, TN 37830, USA

**Keywords:** chronic wasting disease, interspecies transmission, prion, transmissible spongiform encephalopathy, white-tailed deer

## Abstract

The transmission characteristics of prion diseases are influenced by host prion protein sequence and, therefore, the host species. Chronic wasting disease (CWD), a prion disease of cervids, has widespread geographical distribution throughout North America and occurs in both wild and farmed populations. CWD prions contaminate the environment through scattered excrement and decomposing carcasses. Fresh carcasses with CWD prions are accessible by free-ranging mesopredators such as raccoons and may provide a route of exposure. Previous studies demonstrated the susceptibility of raccoons to CWD from white-tailed deer. In this study, we demonstrate that white-tailed deer replicate raccoon-passaged CWD prions which results in clinical disease similar to intraspecies CWD transmission. Six white-tailed deer were oronasally inoculated with brain homogenate from a raccoon with CWD. All six deer developed clinical disease, had widespread lymphoid distribution of misfolded CWD prions (PrP^Sc^), and had neuropathologic lesions with PrP^Sc^ accumulation in the brain. The presence of PrP^Sc^ was confirmed by immunohistochemistry, enzyme-linked immunoassay, and western blot. The western blot migration pattern of raccoon-passaged CWD was different from white-tailed deer CWD. Transmission of raccoon CWD back to white-tailed deer resulted in an interposed molecular phenotype that was measurably different from white-tailed deer CWD.

## 1. Introduction

Transmissible spongiform encephalopathies (TSEs) are a group of neurodegenerative diseases also known as prion diseases [1]. TSEs are caused by a misfolded form (PrP^Sc^) of the normal cellular prion protein [2]. Chronic wasting disease (CWD) is a transmissible spongiform encephalopathy that occurs in cervids. Affected animals succumb to a fatal neurodegenerative process that impairs proprioception, fear responses, and motor function deficits that manifest clinically as emaciation, polyuria, polydipsia, bruxism, excessive salivation, regurgitation of rumen contents, esophageal dilation, and ataxia [3]. Since CWD was recognized in mule deer in Colorado in 1967, it has been identified in free-ranging and commercially farmed cervids spanning 30 US states and 4 Canadian provinces in North America [4].

Infectious CWD prions are spread through direct contact of cervids via exposure to bodily fluids and interactions with infectious prions that remain stable for long periods of time in the environment [5,6,7,8]. The increase in prevalence of CWD has raised concerns about possible transmission to other species [9]. A naturally occurring connection between CWD in cervids and a human prion disease has not been demonstrated to date, and experimental studies that directly inoculated CWD isolates into non-human primates or humanized transgenic mice demonstrate variable results (reviewed by Otero, et al.) [10]. Recently, the susceptibility of humans to CWD was implicated by in vitro experimental work that demonstrated normal human prion protein can misfold when triggered by elk CWD prions and is infectious to humanized transgenic mice [11].

In North America, numerous species are sympatric with farmed and wild cervids. Free-ranging mesopredators may have contact with infectious CWD prions in cervid carcasses or the environment; therefore, mesopredators could participate in a sylvatic transmission cycle. Previous work on the host range of CWD demonstrated that raccoons were susceptible to white-tailed deer CWD prions after intracranial inoculation [12,13]. The purpose of this work was to evaluate the ability of raccoon-passaged CWD to re-transmit to white-tailed deer.

## 2. Materials and Methods

Animals for this experiment were obtained from a CWD-free breeding herd at the United States Department of Agriculture National Animal Disease Center in Ames, IA. Six male castrated white-tailed deer fawns were used in this study. White-tailed deer were homozygous at prion protein polymorphic sites Q95, G96, and Q226.

The inoculum for this experiment was derived from a raccoon that was challenged intracranially with the CWD agent from a white-tailed deer with a prion protein genotype GS96 [13]. The raccoon developed clinical signs consistent with TSE; the presence of PrP^Sc^ was confirmed by immunohistochemistry, western blot, and enzyme immunoassay. The entire frozen half of the brain from the raccoon was homogenized with phosphate-buffered saline (PBS) as a 10% *w*/*v* solution. Deer were oronasally inoculated by elevating the nose and administering 1 mL (0.1 g) of brain homogenate with a needleless syringe into the nares as previously described [14]. During inoculation, deer swallowed the inoculum after passage of material through the nasopharynx to the oropharynx. Deer were housed indoors in a biosafety level 2 agricultural facility and fed a daily ration of pelleted alfalfa. Deer were monitored twice daily for any clinical maladies or signs of disease consistent with transmissible spongiform encephalopathy. The experimental endpoint was the development of unequivocal signs of neurologic disease or 6 years post-inoculation. Deer were euthanized when clinical neurologic, respiratory, or other untreatable intercurrent disease was noted. Euthanasia was performed by intravenous administration of sodium pentobarbital as per label directions under the direction of an animal resources attending veterinarian.

A post-mortem examination was performed on each animal and a routine set of samples were collected. Duplicate tissues were frozen or fixed in 10% buffered neutral formalin. Fixed tissues were embedded in paraffin wax and sectioned (brain, 4 µm; lymphoid, 3 µm; and other, 5 µm) for microscopic analysis of both hematoxylin- and eosin-stained tissue and PrP^Sc^ specific immunohistochemistry. For microscopic analysis, brain, third eyelid, palatine tonsil, pharyngeal tonsil, lymph nodes (mesenteric, retropharyngeal, prescapular, and popliteal), spleen, forestomaches, intestines, and rectal mucosa were assessed. For immunohistochemistry, PrP^Sc^ was labeled with F99 antibody (Anti-Prion Research Kit; Roche Diagnostics, Indianapolis, IN, USA) using an Ventana Discovery XT autostainer (Roche Diagnostics, Indianapolis, IN, USA). Brain tissue from single wild-type prion protein genotype deer inoculated with white-tailed deer CWD was used to compare neuropathology and immunolabeling with raccoon-passaged CWD in deer.

Frozen portions of retropharyngeal lymph node, palatine tonsil, and brainstem at the level of the obex were homogenized using a BeadMill 24 (Fisher Scientific Co., Pittsburgh, PA, USA) and tested for PrP^Sc^ with a commercially available enzyme immunoassay (HerdChek; IDEXX Laboratories, Westbrook, ME, USA) according to kit instructions. Brainstem from the level of the obex was also used to perform western blot analysis. For western blots, samples were initially homogenized at 20% *w*/*v* in PBS. Equal volumes of 20% brain homogenate and RIPA buffer were combined, and the solution was subjected to an additional round of homogenization. Samples were enriched by differential centrifugation with sodium *N*-lauroylsarcosinate solution (sarcosyl). Equal volumes of homogenate and sarcosyl were centrifuged at 10,000× *g* for 30 min at 10 °C. The sarcosyl-soluble supernatant was removed and centrifuged at 186,000× *g* for 55 min at 10 °C. The sarcosyl-insoluble pellet was washed with water and treated with proteinase K at 37 °C for 30 min with agitation. Pefabloc was used to stop the reaction and the solution was centrifuged at 186,000× *g* for 55 min at 10 °C. The resulting pellet was collected and resuspended to 1 mg/uL. Gel electrophoresis and immunodetection of PrP^Sc^ was performed as previously described [15]. Briefly, the samples were combined with β-mercaptoethanol and SDS loading buffer, then heated at 100 °C for 5 min. Gels were run at 200 V for 50 min in 1× MOPS running buffer. Proteins were transferred to low-fluorescence PVDF membrane in a 10% methanol transfer buffer for 1 h at a constant 25 V. The membrane was blocked for 30 min with 3% BSA in TBS with 0.05% Tween-20. Signal detection was achieved using anti-PrP monoclonal antibody SHA31 (Bertin Technologies, Montigny-le-Bretonneux, France) at a dilution of 1:10,000. The primary antibody was incubated at 4 °C overnight. Secondary incubation was performed with a biotinylated sheep anti-mouse IgG secondary antibody (GE Healthcare UK Limited, Amersham, Buckinghamshire, UK) at a dilution of 1:400, followed by streptavidin-HRP (GE Healthcare UK Limited, Amersham, Buckinghamshire, UK) at a dilution of 1:10,000. Both were incubated at room temperature for 1 h. ECL plus detection (ThermoFisher, Thermo scientific, Rockford, IL, USA) was used in conjunction with the iBright visualization system (ThermoFisher, Invitrogen, Waltham, MA, USA) for imaging.

### Ethics Statement

The laboratory and animal experiments were conducted in biosafety level 2 spaces that were inspected and approved for importing prion agents by the US Department of Agriculture, Animal and Plant Health Inspection Service, Veterinary Services. The studies were done in accordance with the Guide for the Care and Use of Laboratory Animals (Institute of Laboratory Animal Resources, National Academy of Sciences, Washington, DC, USA) and the Guide for the Care and Use of Agricultural Animals in Research and Teaching (Federation of Animal Science Societies, Champaign, IL, USA). The protocols were approved by the Institutional Animal Care and Use Committee at the National Animal Disease Center (protocol number: ARS 2018-748), which requires species-specific training in animal care for all staff handling animals.

## 3. Results

All six white-tailed deer inoculated with raccoon-passaged CWD developed clinical disease consistent with CWD. The deer displayed varying symptoms including rough hair coat, cachexia, bruxism, regurgitation of rumen contents, hunched stance, lethargy, circling, and head tremors. Gross lesions at necropsy included ascites, hydropericardium, and serous atrophy of fat. Two of the deer with end-stage CWD developed aspiration pneumonia. The average incubation period was 40 months and ranged from 19 to 62 months post-inoculation (Table 1).

All white-tailed deer had positive EIA results from the brainstem at the level of the obex and the retropharyngeal lymph node. Except for a single deer, #1564, the palatine tonsil was also positive by EIA. Immunohistochemistry demonstrated widespread lymphoid distribution of misfolded prion protein (PrP^Sc^). PrP^Sc^ was variably found within the mesenteric, popliteal, prescapular, and retropharyngeal lymph nodes, small intestine, palatine tonsil, pharyngeal tonsil, rectal mucosal lymphoid tissue, spleen, and lymphoid aggregates of the nictitating membrane. Lymphoid cells in the nictitating membrane were only positive in a single deer, #1558; this deer had the most widespread distribution of PrP^Sc^, as it was demonstrable in all assessed lymphoid tissue. Interestingly, in deer with excellent nictating membrane sample quality (plentiful lymphoid aggregates), PrP^Sc^ was not detected despite abundant immunolabeling in nearby lymphoid tissue (Figure 1). In the forestomaches, small intestine, and cecum, immunolabeling of PrP^Sc^ was observed in the enteric nervous system and the gastrointestinal lymphoid tissue (Figure 2).

To gauge the relative accumulation of PrP^Sc^ throughout the brain of each individual deer, the degree of immunolabeling was compared semi-quantitatively (Figure 3).

The densest areas of immunolabeling were in the olfactory bulb, olfactory tract, and piriform cortex at the level of the basal nuclei (Figure 4), and the brainstem at the level of the obex (Table 2).

The thalamus and hypothalamus contained moderate to heavy aggregates of PrP^Sc^. The least immunolabelling was present in the cerebellum and the neocortex of the cerebrum. The most common immunolabeling types were plaque-like, perivascular, and granular. In multiple deer, the thalamus and midbrain contained multifocal regions of rarefaction of the neuropil and necrosis that contained numerous gitter cells surrounded by large dense aggregates of PrP^Sc^ (Figure 5). Some gitter cells contained immunolabeling for PrP^Sc^. Small- and medium-sized arteries in the vicinity had reactive hypertrophied endothelium and hypertrophied smooth muscle. Deer #1555 had subpial corpora amylacea associated with blood vessels in the brainstem at the level of the obex (Figure 6). The corpora amylacea were not immunoreactive with anti-prion protein antibody.

Small congophilic plaques that demonstrated apple-green birefringence under polarizing light were infrequently observed in some deer brains. Congophilic plaques reacted with anti-prion antibody.

Immunolabeling of PrP^Sc^ in brains from white-tailed deer with raccoon-passaged CWD was compared to PrP^Sc^ from a deer with white-tailed deer CWD. The spongiform change and immunostaining patterns were similar in both types of CWD. Perivascular plaques were the predominate immunolabeling type. The distribution and relative density of PrP^Sc^ throughout each brain region mirrored the brain from deer with raccoon-passaged CWD; although, the overall size of plaques was larger in the white-tailed deer CWD-affected brain.

The western blot of raccoon CWD inoculum compared to CWD from white-tailed deer and deer with raccoon-passaged CWD showed differences between PrP^Sc^ bands (Figure 7). CWD from white-tailed deer had higher banding that was most evident in the unglycosylated PrP^Sc^ fragment. Passage of CWD through raccoons resulted in a lower molecular weight. Transmission of raccoon CWD back to white-tailed deer led to an interposed molecular weight of the unglycosylated fragment in deer 1542, 1546, 1555, and 1558 (Appendix A). The molecular phenotype of deer 1561 and 1564 was more similar to CWD from white-tailed deer than raccoon-passaged CWD. To confirm these results, we performed another western blot of raccoon-passaged CWD samples interspersed with CWD from white-tailed deer (Appendix A).

## 4. Discussion

This study demonstrates that the raccoon-passaged CWD agent readily transmits back to white-tailed deer. Overall, the immunohistochemical and pathologic phenotype of raccoon CWD in white-tailed deer was similar to CWD in white-tailed deer; however, we observed differences in the incubation period and molecular profile characterized by western blot.

Compared to intraspecies passage of CWD in the native host with a wild-type GG96 prion protein genotype, the incubation period was prolonged. The incubation period for matched inoculum to recipient white-tailed deer with the GG96 genotype is around 22 months, and heterologous transmission using GG96 inoculum in GS96 recipient deer is near 31 months [16]. We observed a wide range in incubation period from 19 to 57 months with an average of 40 months. Some variability in this range could be accounted for by a heterogenous mixture of the inoculum, deer coughing up portions of inoculum (i.e., lower infectious dose received), or polygenetic effects on susceptibility [17]. The original white-tailed deer inoculum for this study came from a single deer from Wisconsin that was GS at codon 96. Given that the *PRNP* sequence influences transmission efficiency to other species [18,19], it is probable that raccoon CWD is de-adapted for retro-transmission back to white-tailed deer based on the primary sequence differences between racoons and deer. This is supported by the variable attack rates, (1/4) [12] and 3/5 [13], that demonstrate incomplete transmission after intracranial inoculation.

The clinical, gross pathologic, and histopathologic findings in white-tailed deer with raccoon CWD were similar to white-tailed deer CWD [3,20]. The widespread lymphoid distribution of raccoon CWD was also retained. The finding of vascular-associated concentric laminated bodies was interesting. They were designated as corpora amylacea due to their staining with periodic acid–Schiff and lack of immunoreactivity for PrP^Sc^. Corpora amylacea are associated with normal aging and neurodegenerative diseases in humans; they arise in periventricular, subpial, or perivascular areas [21]. Corpora amylacea were overserved in WTD 1555; this deer was about 5 ½ years old at the time of necropsy.

The finding of multifocal areas of necrosis was unique from other cases of CWD. These lesions could be of a vascular origin or aberrant parasitic migration. White-tailed deer are the definitive host of the meningeal worm, *Parelaphostrongylus tenuis*. Typically, white-tailed deer are asymptomatic hosts unless the parasite burden is high, or the host is debilitated [22]. No nematode larvae or eosinophils were observed, and the lesions were in an atypical location for this parasite in white-tailed deer. Furthermore, being housed indoors, these deer lacked exposure to the intermediate hosts, gastropods. Given that the necrotic areas were vasculocentric and the associated vessels had prominent endothelial cell and smooth muscle hypertrophy, the lesions may have been associated with a vascular ischemic event similar to the pathology of cerebral amyloid angiopathy described in humans [23]. It is unknown if the severity of PrP^Sc^ plaques surrounding these areas precipitated a vascular ischemic event or if the lesion is due to a true vascular leukoencephalopathy.

The finding of gitter cells in an area of encephalomalacia is expected. Gitter cells are CNS-specific phagocytic macrophages derived from blood-borne monocytes and, to a lesser extent, resident microglia. They are involved in the removal of cellular debris, axons, and myelin [24]. Given their function as phagocytic cells and the shear amount of PrP^Sc^ surrounding the lesion, it is not surprising that some gitter cells had intracellular immunolabeling for PrP^Sc^.

The molecular profile of raccoon CWD was different from white-tailed deer CWD. The adaptation of a new molecular phenotype after interspecies prion transmission has been reported [25,26,27]. Interestingly, we observed that passage of raccoon CWD back to white-tailed deer resulted in an interposed molecular phenotype that was measurably different from white-tailed deer CWD. Future work will explore the spectrum of host susceptibility to raccoon-passaged CWD.

## Figures and Tables

**Figure 1 viruses-14-01578-f001:**
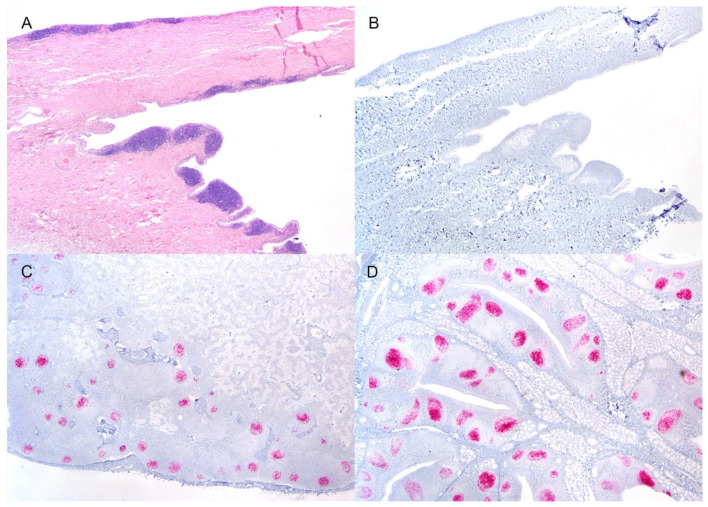
Demonstration of PrP^Sc^ in lymphoid tissue of a white-tailed deer inoculated with raccoon-passaged CWD. (**A**,**B**) Lymphoid aggregates are prominent in the nictitating membrane of this deer; however, no positive immunolabeling is demonstrable. There is abundant immunolabeling of PrP^Sc^ (red) in the retropharyngeal lymph node (**C**) and palatine tonsil (**D**). (**A**) Hematoxylin and eosin. (**B**–**D**) Immunohistochemistry, F99 monoclonal antibody. Original magnification (**A**–**D**) 20×.

**Figure 2 viruses-14-01578-f002:**
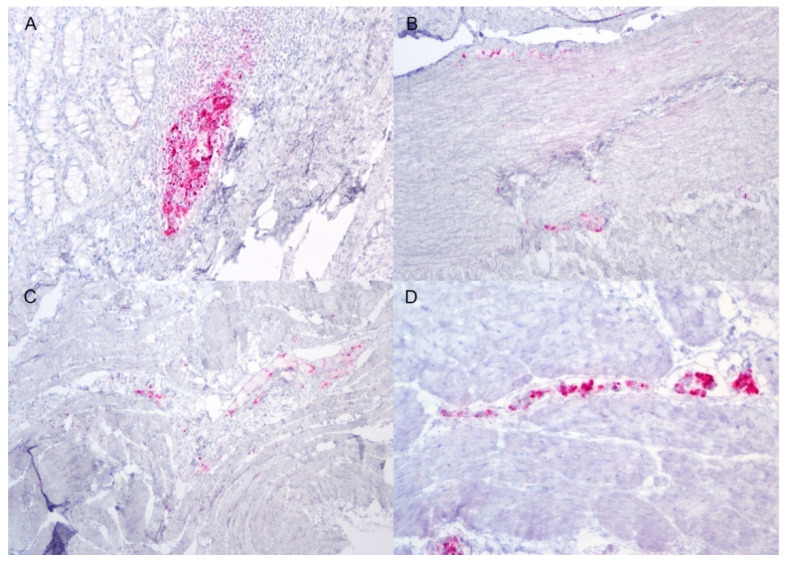
Immunolabeling of PrP^Sc^ in the gastrointestinal tract of white-tailed deer inoculated with raccoon-passaged CWD. There is positive immunolabeling of PrP^Sc^ (red) in the mucosal-associated lymphoid tissue (**A**) and submucosal and myenteric plexuses (**B**) of the small intestine. Immunolabeling is present in the myenteric plexus of the reticulum (**C**) and abomasum (**D**). Immunohistochemistry, F99 monoclonal antibody. Original magnification (**A**–**C**) 100×; (**D**) 200×.

**Figure 3 viruses-14-01578-f003:**
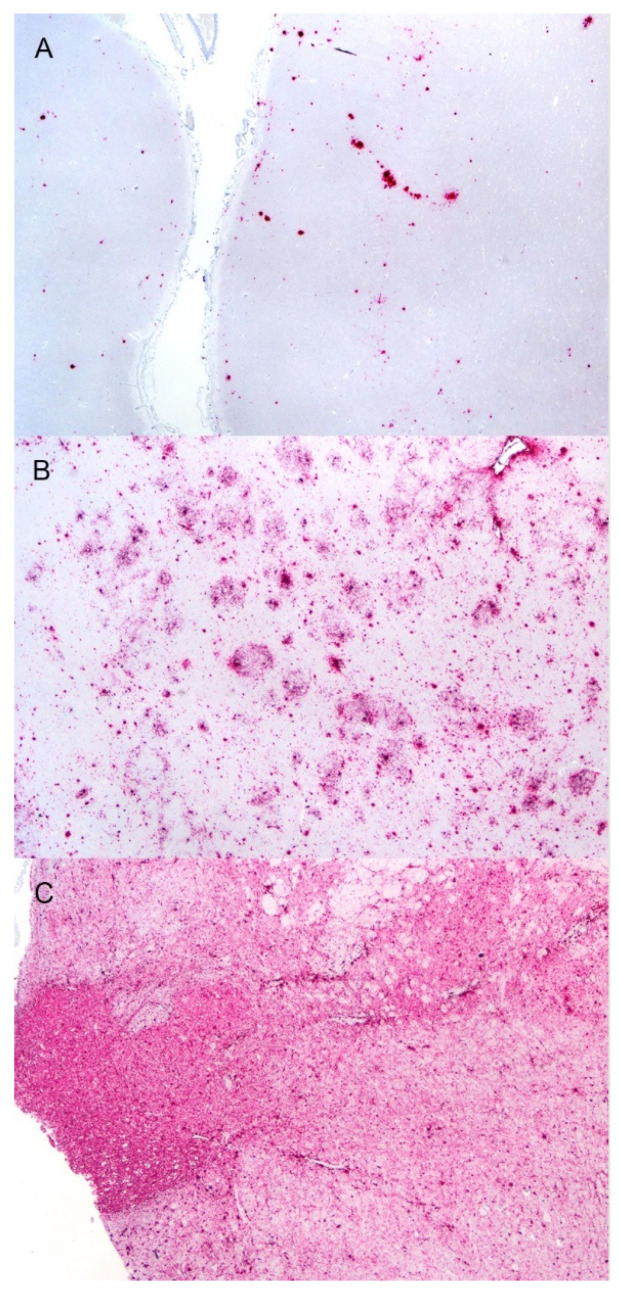
The relative degree of PrP^Sc^ immunolabeling in the brain of deer with raccoon-passaged CWD. (**A**–**C**) Images depict mild (+), moderate (++), and severe (+++) amounts of PrP^Sc^ in the brain. Immunohistochemistry, F99 monoclonal antibody. Original magnification (**A**–**C**) 40×.

**Figure 4 viruses-14-01578-f004:**
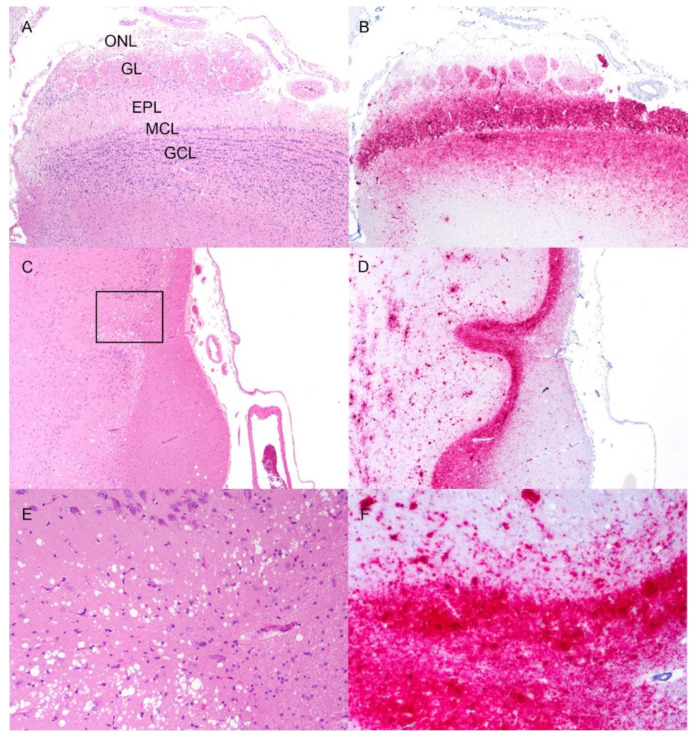
Immunolabeling of PrP^Sc^ in the olfactory pathway of white-tailed deer inoculated with raccoon CWD. (**A**) The layers of the olfactory bulb from rostral to caudal (top of image to bottom) are the olfactory nerve cell layer (ONL), the glomerular layer (GL), the external plexiform layer (EPL), the mitral cell layer (MCL), and the granule cell layer (GCL). Spongiform change is present in the EPL and less notably in the granule cell layer. (**B**) Immunolabeling for PrP^Sc^ (red) is strongest in the EPL and GCL. (**C**,**D**) The optic tract at the level of the basal nuclei has dense accumulation of PrP^Sc^ (red). (**E**,**F**) A higher magnification of the area outlined by the rectangle in (**C**). (**E**) There is severe spongiform change in the piriform cortex. (**F**) There are coalescing granular and plaque-like aggregates of PrP^Sc^ in the olfactory tract (red). (**A**,**C**,**E**) Hematoxylin and eosin. (**B**,**D**,**F**) Immunohistochemistry, F99 monoclonal antibody. Original magnification (**A**–**D**) 20×; (**E**,**F**) 100×.

**Figure 5 viruses-14-01578-f005:**
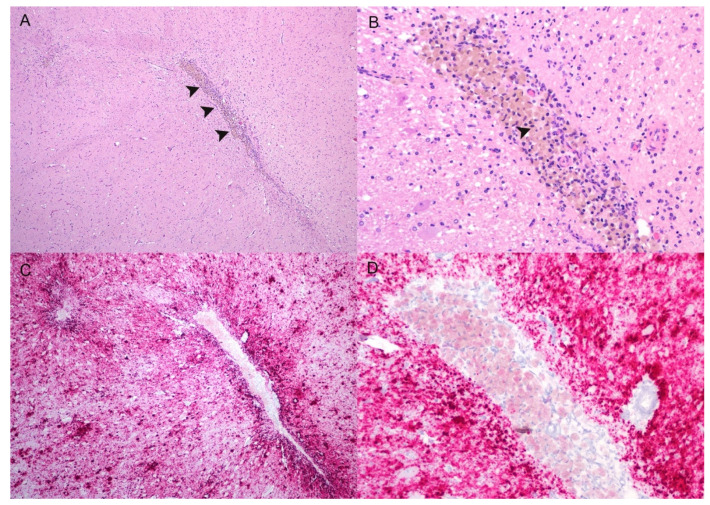
Neuroinflammation and necrosis at the level of the thalamus. (**A**) The neuroparenchyma is disrupted by a linear tract with a cellular infiltrate and necrosis (arrowheads). (**B**) A higher magnification of the same area from panel (**A**). The predominate inflammatory cells are gitter cells that contain golden-brown pigment (arrowhead); an increase in microglia and astrocytes is also present. Small- and medium-sized blood vessels have hypertrophied smooth muscle and endothelium. (**C**,**D**) Dense aggregates of PrP^Sc^ (red) surround the focal areas of necrosis and inflammation. The lesions and PrP^Sc^ immunolabeling are vasculocentric. There is intracytoplasmic PrP^Sc^ within gitter cells. (**A**,**B**) Hematoxylin and eosin. (**C**,**D**) Immunohistochemistry, F99 monoclonal antibody. Original magnification (**A**,**C**) 100×; (**B**,**D**) 200×.

**Figure 6 viruses-14-01578-f006:**
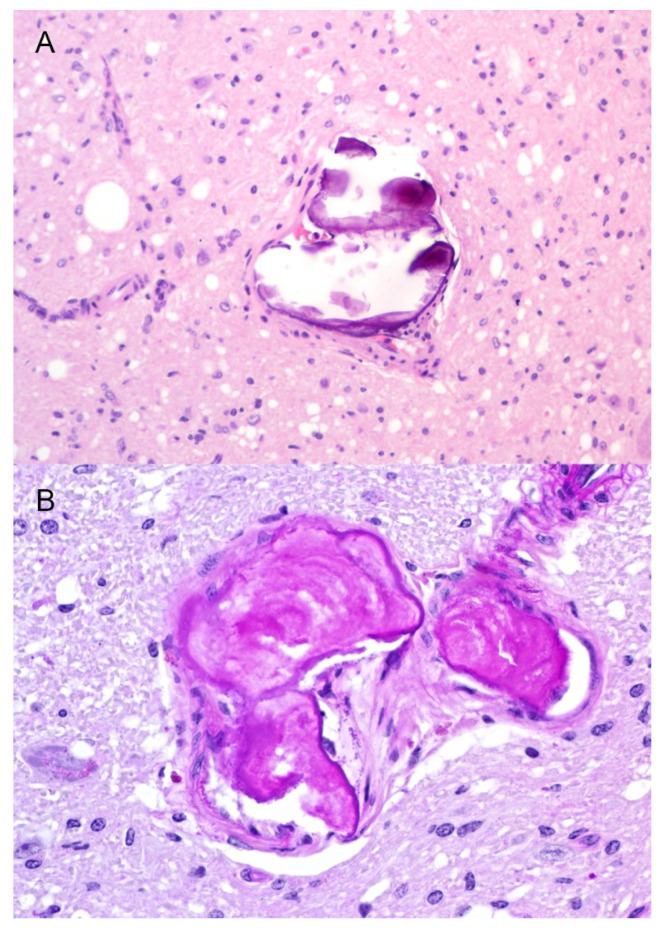
Corpora amylacea at the level of the obex. (**A**) Hematoxylin and eosin stain of calcified material in the subpial layer of a small vessel. (**B**) Periodic acid–Schiff stain is positive (fuchsia/magenta) for glycoproteins in the concentrically laminated material. Original magnification (**A**) 400×; (**B**) 600×.

**Figure 7 viruses-14-01578-f007:**
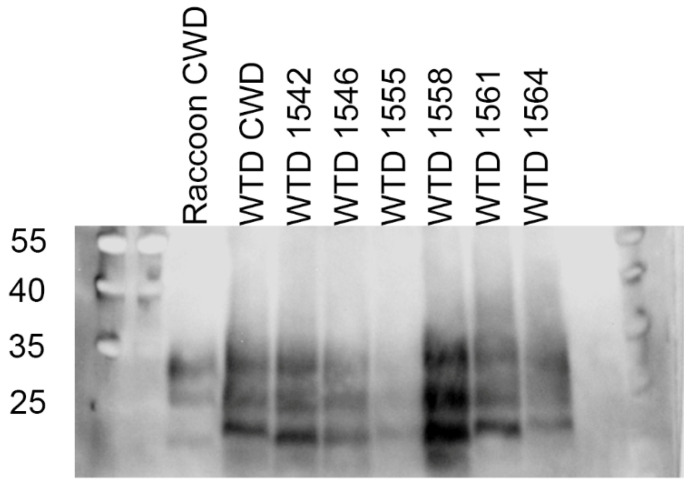
Western blots probing for PrP^Sc^. Raccoon CWD and white-tailed deer inoculated with raccoon CWD (1542, 1546, 1555, 1558, 1561, 1564) had smaller unglycosylated PrP^Sc^ fragments (lower/further migration) compared to chronic wasting disease straight from a GG96 white-tailed deer (WTD CWD). Gel loading tissue equivalents left to right (mg): 8, 1, 1, 2, 3, 2, 1.5, 4. Molecular weights are displayed in kDa. Immunostaining for PrP^Sc^ was performed with monoclonal antibody SHA31.

**Table 1 viruses-14-01578-t001:** Incubation periods and immunodetection of PrP^Sc^ in white-tailed deer oronasally inoculated with raccoon-passaged chronic wasting disease.

		RPLN		Palatine Tonsil		Spleen	Mesenteric Lymph Node	RAMALT
ID #	IP	EIA	IHC	EIA	IHC	IHC	IHC	IHC
1542	34	+	+	+	+	+	+	+
1546	30	+	+	+	+	+	+	+
1555	62	+	+	+	+	nd	+	+
1558	19	+	+	+	+	+	+	+
1561	38	+	+	+	+	+	+	+
1564	57	+	+	nd	+	nd	+	+

RPLN, retropharyngeal lymph node; RAMALT, recto-anal mucosa-associated lymphoid tissue; ID, deer identification #; IP, incubation period (months); EIA, enzyme immunoassay; IHC, immunohistochemistry; nd, not detected.

**Table 2 viruses-14-01578-t002:** Relative amount of PrP^Sc^ throughout the brain in individual white-tailed deer inoculated with raccoon CWD based on immunohistochemistry.

Neuroanatomic Location	1542	1546	1555	1558	1561	1564
Olfactory bulb	+++	+++	+++	n/a	+++	++
Olfactory tract	+++	+++	+++	+++	+++	++
Basal nuclei	++	++	++	+	++	+
Thalamus	+++	+++	++	++	++	++
Hypothalamus	++	++	+++	+	++	++
Colliculi	+++	++	+++	++	+++	+
Pons	+++	++	+++	++	+++	+
Cerebellum	++	+	+	+	++	+
Brainstem, obex	+++	+++	+++	+++	+++	+++
Neocortex, caudate nucleus	++	+	+	+	++	+
Neocortex, thalamus	+	+	+	+	+	+

The degree of immunoreactivity is illustrated in Figure 3 (+ mild, ++ moderate, +++ severe); n/a, not available. Neocortex immunoreactivity was assessed at the levels of the caudate nucleus and thalamus.

## Data Availability

The data presented in this study are available on request from the corresponding author.

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
