# Peer review of "Transmission of Raccoon-Passaged Chronic Wasting Disease Agent to White-Tailed Deer"

_viruses, 2022, doi:10.3390/v14071578_

Round 1

Reviewer 1 Report

This is a well-written manuscript describing the incubation period and pathology of racoon passaged CWD back into white-tailed deer. The introduction and methods are clear, concise and provide the reader with the information needed to understand the significance of the results.  The results are of interest to the transmission properties of CWD but could be presented more clearly, especially the photomicrographs and the western blot.

Major concerns:

  1. All photomicrographs should include a scale bar for reference and the multiple panels should be separated by either white space or black lines to make them easier to interpret.
  2. Some of the figures would be improved by labeling of structures within the panel (for example the labeling of the optic tract in Figure 4 C and D would be helpful). Also, it would be helpful to provide the reader with more information about adjacent panels (are Panels E and F in Figure 4 adjacent to each other, or nearly adjacent?; and in Figure 5 what is the relationship between the four panels?).
  3. In figure 5 the area of necrosis sure looks like a needle track, but since the animals were not inoculated IC then there is no reason for a needle track, correct? And there should be a high power view of a gitter cell and some discussion of these cells since this is the first report that I am aware of that demonstrates that gitter cells accumulate PrPsc.  The arrow in panel B points to something that cannot be seen at the current magnification.  Are gitter cells activated microglia?  This seems like a significant finding that could be expanded upon.
  4. The western blot data, which is very important, would benefit from improved presentation.  Would it be possible to add additional WTD CWD lanes so that it is easier to compare it to the lanes on the right? It is very difficult to be certain that the lower molecular weight band is different from the lanes on the right side of the figure.  The change in PrPSc fragments following passage through racoons has potential implications for the zoonotic potential of CWD (especially since the route of inoculation in this study is a potentially natural route) and it would be easier to assess this data with a clearer presentation of the western blot.  Some mention of the zoonotic significance of these findings should be made in the discussion.
  5. Table 1. Consider changing nd to (-) since you use the (+) designation and "nd" is often assumed to be "not done" instead of not detected. There is RAMALT abbreviation in the caption, but no RAMALT in the table.
  6. Table 2.  Is "Olfactory Tract" really the olfactory tract, or is it the olfactory bulb as shown in Figure 4A?  Is "Colliculus" referring to the inferior colliculus, the superior colliculus, or both colliculi? It is not clear what is meant by "Neocortex, caudate nuclei"? Why is caudate plural and is this tissue that includes both structures, or is it cortex adjacent to/or at the level of the caudate? Same for "Neocortex, thalamus".
  7. Typographical errors:
    1. Line 81 "and" should be "or".
    2. Line 98 "subjugated" should be "subjected".
    3. Line 150 "variably" should be clarified. Was PrPsc found in all of these structures from all of the animals that they were collected from or were they found in only some of the tissues from all of the animals they were collected from?
    4. Figure 4 "olfactory tract" should be changed to "olfactory bulb".
    5. Line 252 add "and" between "period" and "molecular".
    6. Line 272 "reaction" should be changed to "staining".

Author Response

Reviewer 1 

Comments and Suggestions for Authors 

This is a well-written manuscript describing the incubation period and pathology of racoon passaged CWD back into white-tailed deer. The introduction and methods are clear, concise and provide the reader with the information needed to understand the significance of the results.  The results are of interest to the transmission properties of CWD but could be presented more clearly, especially the photomicrographs and the western blot. 

Thank you for your comments. We addressed the concerns and made changes where appropriate. In places where we did not alter the manuscript, we provided justification for our decision.  

Major concerns: 

  1. All photomicrographs should include a scale bar for reference and the multiple panels should be separated by either white space or black lines to make them easier to interpret. 

We followed the guidelines for authors that were specific for this journal (Viruses | Instructions for Authors (mdpi.com)). Furthermore, as it relates to scale bars in veterinary pathology specific journals, it’s not recommended. The journal Veterinary Pathology instructs authors as follows  

Do not use scale bars in gross or histologic images, except in rare cases where their use is critical to the understanding of the image; justification for their use must be included in the cover letter. 

No changes were made. 

  1. Some of the figures would be improved by labeling of structures within the panel (for example the labeling of the optic tract in Figure 4 C and D would be helpful).  

The optic tract is defined and outlined in the figure legend as an area of “dense accumulation of PrPSc (red).” It is clearly seen without extra arrows.  

Also, it would be helpful to provide the reader with more information about adjacent panels (are Panels E and F in Figure 4 adjacent to each other, or nearly adjacent?;  

Panels E and F are defined in the legend. “(E, F) A higher magnification of the area outlined by the rectangle in (C).” 

and in Figure 5 what is the relationship between the four panels?). 

For panel B, we added the following text to the legend “A higher magnification of the same area from A”. The legend already describes the relationship between A/B and C/D as Hematoxylin & Eosin staining  vs IHC.  

  1. In figure 5 the area of necrosis sure looks like a needle track, but since the animals were not inoculated IC then there is no reason for a needle track, correct?  

Correct, these deer were not intracranially inoculated and did not have iatrogenic needle tracts. 

And there should be a high power view of a gitter cell and some discussion of these cells since this is the first report that I am aware of that demonstrates that gitter cells accumulate PrPsc.  The arrow in panel B points to something that cannot be seen at the current magnification.   

The arrow points at a gitter cell, and that is visible at this magnification.  

Are gitter cells activated microglia?  This seems like a significant finding that could be expanded upon. 

We added the following comments to the discussion 

The finding of gitter cells in an area of encephalomalacia is expected. Gitter cells are CNS specific phagocytic macrophages derived from blood borne monocytes and, to a lesser extent, resident microglia. They are involved in the removal of cellular debris, axons, and myelin [24]. Given their function as phagocytic cells and the shear amount of PrPSc surrounding the lesion, it’s not surprising that some gitter cells had intracellular immunolabeling for PrPSc. 

  1. The western blot data, which is very important, would benefit from improved presentation.  Would it be possible to add additional WTD CWD lanes so that it is easier to compare it to the lanes on the right? It is very difficult to be certain that the lower molecular weight band is different from the lanes on the right side of the figure.  The change in PrPSc fragments following passage through racoons has potential implications for the zoonotic potential of CWD (especially since the route of inoculation in this study is a potentially natural route) and it would be easier to assess this data with a clearer presentation of the western blot.  Some mention of the zoonotic significance of these findings should be made in the discussion. 

Thank you, we agree with the reviewer’s comments here. Unfortunately, we are unable to run further western blots because of exhausted Raccoon CWD sample. In the western blot presented in the manuscript, the relative band front varies between each side of the gel. Due to this and the reviewers concern, we calculated the actual molecular weights and presented the information I the manuscript. The additional analysis was included as Supplementary file 1.  

We are unable to speculate about zoonotic potential of these isolates currently. However, we agree that the implications of a molecular shift are significant. We tried to capture this importance in the manuscript’s last sentence. 

  1. Table 1. Consider changing nd to (-) since you use the (+) designation and "nd" is often assumed to be "not done" instead of not detected. There is RAMALT abbreviation in the caption, but no RAMALT in the table. 

Thank you for this comment. The use of negative (by elisa) can be misleading because bioassays and amplification assays demonstrate positivity in “negative by elisa” samples. Also, “non detect” is the designation applied in TSE surveillance and diagnosis by national veterinary services laboratory. For clarification, we defined nd as “not detected” in the table footnote. 

RAMALT is the last column to right. The table might not be displaying fully on your screen. Try viewing the document in WebLayout if Viruses allows you to download a word file.  

  1. Table 2.  Is "Olfactory Tract" really the olfactory tract, or is it the olfactory bulb as shown in Figure 4A?   

Thank you for your comment and catching this typo. cortex was changed to bulb in table 2.  

Is "Colliculus" referring to the inferior colliculus, the superior colliculus, or both colliculi?  

both 

It is not clear what is meant by "Neocortex, caudate nuclei"? Why is caudate plural and is this tissue that includes both structures, or is it cortex adjacent to/or at the level of the caudate? Same for "Neocortex, thalamus". 

Since we only looked at slides from one side of the brain, nuclei was changed to nucleus. Neocortex at the level of the caudate nucleus is too big to fit in the column so we shortened it to “neocortex, caudate nuclei. To clarify, we explained the abbreviation in the table footnote.  

  1. Typographical errors: 
  1. Line 81 "and" should be "or". 

Done 

  1. Line 98 "subjugated" should be "subjected". 

Done 

  1. Line 150 "variably" should be clarified. Was PrPsc found in all of these structures from all of the animals that they were collected from or were they found in only some of the tissues from all of the animals they were collected from? 

The next sentence says, “Lymphoid cells in the nictating membrane were only positive in a single deer, #1558.” This means that “variably” signifies PrPSc was “found in only some of the tissues from all of the animals they were collected from” 

  1. Figure 4 "olfactory tract" should be changed to "olfactory bulb". 

Thank you for noticing this. We changed tract to pathway and clarified whether we were referring to bulb, tract, or piriform cortex. 

  1. Line 252 add "and" between "period" and "molecular". 

Done 

  1. Line 272 "reaction" should be changed to "staining". 

done 

Thank you. All suggestions incorporated. 

Reviewer 2 Report

This is a straight forward study to evaluate whether raccoon passaged CWD can be transmitted back to white-tailed deer. Given the wide spread CWD in free ranging animals North America, the significance of this study is high. The approach is appropriate and results are analyzed properly. The finding that raccoon passaged CWD can be readily transmitted to white-tailed deer with a 100% attack rate is quite interesting. The following points need to be addressed to improve the manuscript. 

  1. The sentence "The glycoform migration ..." in the abstract (line22-23) is not very clear. Are authors talking about the ratio of different glycoforms or the migration of differently glycosylated bands.
  2. The sentence "...resulted in a lower molecular weight that was maintained after transmission back..."(line238-239) is not entirely accurate. From figure 7, it appears that some of the deers like #1561 and #1564 are similar to that of CWD in white-tailed deer. #1555 appears to be in between. More detailed description and/or some discussion may help readers. 
  3. Because part of the table 2 was missing from the pdf file, it is not clear whether the PrPSc distribution pattern of CWD in white-tailed deer was included. If not, it should be included to allow readers to compare. 
  4. One of the interesting point is to compare raccoon passaged CWD in deer to that of CWD in deer. However, the paragraph from line 228 to 234 describing this comparison could be improved. It is not clear wether the sentence "Perivascular..."(line230-231) refers to CWD in deer or raccoon passaged CWD in deer. Table 2 could be cited after the next sentence. Examples of the larger plaques in white-tailed deer CWD affected brain could be shown. 
  5. Is the inocula for raccon and the CWD in white-tailed deer from the same isolate? This information should be included.
  6. Typo in line 153 "nictating"

Author Response

Reviewer 2 

Comments and Suggestions for Authors 

This is a straight forward study to evaluate whether raccoon passaged CWD can be transmitted back to white-tailed deer. Given the wide spread CWD in free ranging animals North America, the significance of this study is high. The approach is appropriate and results are analyzed properly. The finding that raccoon passaged CWD can be readily transmitted to white-tailed deer with a 100% attack rate is quite interesting. The following points need to be addressed to improve the manuscript.  

  1. The sentence "The glycoform migration ..." in the abstract (line22-23) is not very clear. Are authors talking about the ratio of different glycoforms or the migration of differently glycosylated bands. 

We adjusted the abstract to clarify. 

  1. The sentence "...resulted in a lower molecular weight that was maintained after transmission back..."(line238-239) is not entirely accurate. From figure 7, it appears that some of the deers like #1561 and #1564 are similar to that of CWD in white-tailed deer. #1555 appears to be in between. More detailed description and/or some discussion may help readers.  

We clarified and provided a new supplementary file to help with analysis.  

  1. Because part of the table 2 was missing from the pdf file, it is not clear whether the PrPSc distribution pattern of CWD in white-tailed deer was included. If not, it should be included to allow readers to compare.  

We included results of PrPSc distribution from CWD in WTD in the text.  

“The distribution and relative density of PrPSc throughout each brain region mirrored the brain from deer with raccoon passaged CWD” 

  1. One of the interesting point is to compare raccoon passaged CWD in deer to that of CWD in deer. However, the paragraph from line 228 to 234 describing this comparison could be improved. It is not clear wether the sentence "Perivascular..."(line230-231) refers to CWD in deer or raccoon passaged CWD in deer. Table 2 could be cited after the next sentence. Examples of the larger plaques in white-tailed deer CWD affected brain could be shown.  

The size of plaques could be a function of disease timepoint not necessarily strain characteristics, so we chose not to make this a defining point in the paper. For this same reason, it’s important to only compare the degree of immunoreactivity between different brain regions “within” a single animal’s brain and not between animals, because there is variability in immunoreactivity based on incubation periods. This is why we titled table 2 as “Relative amount of PrPSc throughout the brain in individual white-tailed deer” 

  1. Is the inocula for raccon and the CWD in white-tailed deer from the same isolate? This information should be included. 

The isolate is not the same. The information is in the methods.  

  1. Typo in line 153 "nictating" 

Corrected 

Reviewer 3 Report

Summary:

The authors present a study focusing on chronic wasting disease (CWD) and its potential to cross species. In the article show and characterized the susceptibility of white tail deer to be infected by CWD passaged into raccoons. The study is overall interesting and well written. Nevertheless, there are important issues with both figures and tables appearing truncated in the pdf. In addition, the last argument regarding the PrPSc lower band differences is far from convincing.

Major comments:

  • Table1: RAMALT (recto-anal mucosa associated lymphoid tissue) columns are missing from table 1.

  • Immunochemistry pictures need scale bars. Authors should also report which objectives magnification were used. Overall, pictures are of good quality, but it would be helpful to readers to use arrowheads (in the like to figure 5) and indicate the critical elements reported in the result section.

  • In figures 1, 2, 4 and 5 the right panels (B and D) appear truncated in the pdf with a large section missing. The authors need to correct that.

  • Like the immunochemistry panels, both tables 1 and 2 are also truncated in the pdf and need fixing.

  • Figure 7: Overall the western blot and the argument made lines 235 to 240 are not very convincing. I would argue that 3 out of 6 deer (1542, 1546 and 1555) show the phenotype proposed by the authors. However, the other 3 (1558, 1561 and 1564) are the complete opposite and have their lower bands arguably higher than the wild type WTD CWD. Salt and protein concentration can impact how proteins will run/appear in a western blot and I am worried that the phenotype argued here may be an artefact, in particular due to the large discrepancies of tissue inputs needed (8 to 1). To really prove there is a consequential band change that is maintained, authors would probably need to run mass spectrometry.

Minor comments:

  • Line 11: TSE should be defined here as it is its first use.

  • Line 27-28: Authors forgot to remove part of the template (List three to ten pertinent keywords specific to the article yet reasonably common within the subject discipline).

  • Material and methods are overall fine but they should be broken down in specific sections with headers to make it easier on the reader to spot the correct information.

Author Response

Reviewer 3 

Summary: 

The authors present a study focusing on chronic wasting disease (CWD) and its potential to cross species. In the article show and characterized the susceptibility of white tail deer to be infected by CWD passaged into raccoons. The study is overall interesting and well written. Nevertheless, there are important issues with both figures and tables appearing truncated in the pdf. In addition, the last argument regarding the PrPSc lower band differences is far from convincing. 

  

Major comments: 

  • Table1: RAMALT (recto-anal mucosa associated lymphoid tissue) columns are missing from table 1. 

 This is a biproduct of placing the table in the preformated manuscript template. If we are unable to remediate this, it may require the assistance of a help from an assistant editor. 

  • Immunochemistry pictures need scale bars. Authors should also report which objectives magnification were used. Overall, pictures are of good quality, but it would be helpful to readers to use arrowheads (in the like to figure 5) and indicate the critical elements reported in the result section. 

 We followed the guidelines for authors that were specific for this journal (Viruses | Instructions for Authors (mdpi.com)). Furthermore, as it relates to scale bars in veterinary pathology specific journals, it’s not recommended. The journal Veterinary Pathology instructs authors as follows  

Do not use scale bars in gross or histologic images, except in rare cases where their use is critical to the understanding of the image; justification for their use must be included in the cover letter. 

No changes were made. 

  • In figures 1, 2, 4 and 5 the right panels (B and D) appear truncated in the pdf with a large section missing. The authors need to correct that. 

These figures are not truncated in the WORD file submitted, so must be an element of the PDF conversion process. We will attempt to repair this, but it may be necessary to review these figures in the WORD file. 

  

  • Like the immunochemistry panels, both tables 1 and 2 are also truncated in the pdf and need fixing. 

Thank you. Please see above. 

  

  • Figure 7: Overall the western blot and the argument made lines 235 to 240 are not very convincing. I would argue that 3 out of 6 deer (1542, 1546 and 1555) show the phenotype proposed by the authors. However, the other 3 (1558, 1561 and 1564) are the complete opposite and have their lower bands arguably higher than the wild type WTD CWD. Salt and protein concentration can impact how proteins will run/appear in a western blot and I am worried that the phenotype argued here may be an artefact, in particular due to the large discrepancies of tissue inputs needed (8 to 1). To really prove there is a consequential band change that is maintained, authors would probably need to run mass spectrometry. 

Thank you, we agree with the reviewer’s comments here. Unfortunately, we are unable to run further western blots because of exhausted Raccoon CWD sample. In the western blot presented in the manuscript, the relative band front varies between each side of the gel. Due to this and the reviewers concern, we calculated the actual molecular weights and presented the information I the manuscript. The additional analysis was included as Supplementary file 1.  

Minor comments: 

  • Line 11: TSE should be defined here as it is its first use. 

Changed to prion disease to avoid TSE. 

  • Line 27-28: Authors forgot to remove part of the template (List three to ten pertinent keywords specific to the article yet reasonably common within the subject discipline). 

  Corrected 

  • Material and methods are overall fine but they should be broken down in specific sections with headers to make it easier on the reader to spot the correct information. 

 The materials and methods are prepared according to the instruction for authors. No changes made. 

Round 2

Reviewer 3 Report

The authors made minimal efforts to address my comments. In particular, the Figure 7 and the conclusions made by the authors are not convincing enough.

As I mentioned in the previous rebuttal " I would argue that 3 out of 6 deer (1542, 1546 and 1555) show the phenotype proposed by the authors. However, the other 3 (1558, 1561 and 1564) are the complete opposite and have their lower bands arguably higher than the wild type WTD CWD." That was not addressed.

Furthermore, the "analysis" presented in figure 1 is extremely biais and do not prove anything really. It is bias because the boundaries of the areas are selected differently in favor of the authors arguments (different sizes and different starting point of what is considered the "band"). If authors had presented a second protein (any internal control really), perform the same analysis on it and shown no difference then that analysis could have provided some input. However in its current form, it does not.

I understand that the authors ran out of protein sample but how?

- They only present one western blot membranes. Did they ran the samples multiple times? Where are these others images? It could have been so easy to demonstrate the data was reproducible at least to present it more convincingly.

- Why would they not strip the membrane and probe it with any internal control to show the samples ran similarly across? It is an obvious and minimal process to perform and provide reliability. As mentioned previously samples can ran differently and present artifact especially due to the large discrepancies of tissue inputs needed (8 to 1 here).

Overall, the claim put forward by the authors is not supported convincingly by the data presented.

Author Response

The western blot reported as figure 7 was repeated with similar results submitted as supplemental figure 2. The original raccoon sample was not available for comparison since it was used to conduct multiple other interspecies transmission experiments and the original half brain available was less than 15 g. We disagree that the migration differences presented in figure 7 represent an artifact as samples were treated by the same conditions and are only loaded with different amounts to allow for uniform detection without increased exposure for weak samples making bands from stronger samples blend together.  To imply that the results of the glycoform analysis are due to the authors selecting specific regions to support our claims is absurd. The regions for analysis are selected by the imager software and a relative front is determined based upon the markers. This allows for calculation of relative molecular weights even if the blot is imperfect. Refinement is only done by the user if the lane profile analysis shows that the pixel intensities are not strongest in the region that is autoselected.